# The impact of chewing khat during pregnancy on selected pregnancy outcomes in eastern Ethiopia: A cohort study with a generalized structural equation modeling analysis approach

**Amsalu Taye Wondemagegn**[1,2]*, **Miressa Bekana**[3], **Yonas Bekuretsion**[4], **Mekbeb Afework**[1]

1 Department of Anatomy, School of Medicine, College of Health Sciences, Addis Ababa University, Addis Ababa, Ethiopia, 2 Department of Biomedical Sciences, School of Medicine, Debre Markos University, Debre Markos, Ethiopia, 3 Department of Obstetrics and Gynecology, School of Medicine, College of Health Sciences, Haramaya University, Harar, Ethiopia, 4 Department of Pathology, School of Medicine, College of Health Sciences, Addis Ababa University, Addis Ababa, Ethiopia

* 50amsalu@gmail.com

## Abstract

### Introduction

Little is known about the fetal and pregnancy effects of khat chewing during pregnancy. The aim of the current study was to determine the impact of chewing khat during pregnancy on selected pregnancy outcomes in Ethiopia, 2022: A Cohort Study with a Generalized Structural Equation Modeling Analysis Approach.

### Methods

An institution-based prospective cohort study was employed in selected hospitals in eastern Ethiopia. Pregnant women who visited the selected hospitals in the study area during the study period that fulfilled the eligibility criteria were included until a sample size (344) was fulfilled. The main outcomes studied in the present study were preterm birth and low birth weight. Data were collected through anthropometric and clinical measurements, and interviewers administered questionnaires. The survival analysis and generalized linear model analysis were performed to estimate the crude and adjusted relative risk and attributable risk. The Generalized Structural Equation Modeling (GSEM) analysis was performed using the Statistical software for data science (Stata) 'GSEM' command to examine the mediation effect.

### Results

The risk of occurrence of preterm birth was significantly higher among khat chewers [adjusted relative risk (aRR) = 2.19; 95%CI 1.21–3.96]. In further analysis after adjusting for gestational hypertension and cesarean delivery, the regression coefficient of khat chewing

**Data Availability Statement:** All relevant data are within the manuscript and its Supporting Information files.

**Funding:** Financial support for this research was obtained from Addis Ababa University and Debre Markos University.The funders had no role in study design, data collection and analysis, decision to publish, or preparation of the manuscript.

**Competing interests:** The authors have declared that no competing interests exist.

**Abbreviations:** aRR, Adjusted Relative Risk; ANC, Ante Natal Care; AR, Attributable Risk; BMI, Body Mass Index; CI, Confidence Interval; GLM, Generalized Linear Model; GSEM, Generalized Structural Equation Modeling; IQR, Inter Quartile Range; LMP, Last Menstrual Period; LBW, Low Birth Weight; MUAC, Mid-Upper Arm Circumference; PROM, Pre-labor Rupture of Membranes; VLBW, Very Low Birth Weight; SD, Standard deviation; WHO, World Health Organizatio.

during pregnancy on preterm birth has been decreased in size from path n, $\beta = 0.37$, $p<0.001$ to path n', $\beta = 0.15$, $p<0.005$. The risk of occurrence of low birth weight among khat chewers was significantly higher (aRR = 4.17; 95%CI 2.11–8.25). In further analysis after adjusting for gestational hypertension, cesarean delivery, preterm birth and maternal anemia, the regression coefficient of khat chewing during pregnancy on low birth weight has been decreased in size from path q, $\beta = 0.4$, $p<0.001$ to path q', $\beta = 0.2$, $p<0.001$.

## Conclusion

Overall, the present study revealed that khat chewing is not only a worry of the current population but also a public health concern of the generation affecting unborn fetuses.

## Introduction

Antenatal substance use is related with numerous harmful pregnancy and fetal effects. Alcohol use, smoking, cocaine use, cannabis use, and other substance use in pregnancy have been associated with a range of negative maternal and birth outcomes [1–5]. In the same way, according to a few available animal experimental studies, khat use during pregnancy is associated with adverse pregnancy and fetal outcomes [6–8]. There are also a few human studies that have revealed the association between prenatal khat exposure and fetal outcomes. A cross-sectional study [9] conducted in Yemen reported that khat chewing mothers were shown to give birth to more low birth-weight babies than non-khat-chewers. Another study [10], also conducted in Yemen, reported that khat chewing during pregnancy significantly lowered birth weight. Similarly, a facility-based case-control study [11] in Ethiopia found that maternal khat chewing during pregnancy has been associated with lower birth weight.

Khat encompasses many chemical constituents that will have various effect on the body structures. The main active component of chat responsible for its stimulant effect is a psychoactive alkaloid chemical known as cathinone [12, 13], which is structurally and chemically similar to amphetamine.

Cathinone is an extremely forceful stimulant, which resulted in sympathomimetic and centralized nervous system stimulation and causes release of symptoms like euphoria and hyperactivity following consumption of leaves, which is equivalent to the effect of amphetamine [13]. The results of numerous *in-vivo* and *in-vitro* experiments [14–17] showed that the substance, khat might be considered as a natural amphetamine.

Even though khat is consumed in all corners of the country, it is freely chewed by almost every segment of the population, including pregnant mothers in the eastern parts of Ethiopia. Hence, conducting the current study in this area is more appropriate.

The existing limited human studies reported regarding the association between khat chewing during pregnancy and low birthweight are cross-sectional and case control studies in nature, which are not suitable to establish a temporal relationship, unlike that of cohort study. Moreover, despite the few reports that have existed revealing the association between khat chewing and low birth weight using cross-sectional and case control data, these few studies have not conducted mediation analysis to explain how khat chewing affects low birth weight. Hence, up to the researcher's knowledge, the present study is the first to demonstrate a model by which selected variables (gestational hypertension, maternal anemia, emergency cesarean section delivery, and preterm birth) mediate the association between khat chewing during pregnancy and selected pregnancy outcomes, thereby explaining the mechanism by which

khat chewing during pregnancy can influence selected pregnancy outcomes. In addition, there are limited human studies that have revealed the association between chewing khat during pregnancy and preterm birth and stillbirth in a prospective cohort study approach. Hence, the aim of the current study was to determine the impact of chewing khat during pregnancy on selected pregnancy outcomes in Eastern Ethiopia, 2022: A Cohort Study with a Generalized Structural Equation Modeling Analysis Approach.

## Methods and materials

### Study design and setting

A multi-site prospective longitudinal open cohort study of pregnant women who chewed khat (exposed) and not chewed khat (unexposed) was conducted from August to December 2022 in selected hospitals of Dire Dawa administration, Harari region, and Jigjiga city administration, eastern Ethiopia. The recruitment period was from August 15 to September 15, 2022. The current study compared selected pregnancy outcomes for women with confirmed khat chewing practice during pregnancy (exposed) with pregnant women not practicing khat chewing during the pregnancy period (unexposed). Those participants at high risk of adverse birth outcomes like having known major chronic illness such as diabetes mellitus and cardiovascular diseases; and having previous history of congenital anomalies were excluded. Moreover, those pregnant mothers with multiple pregnancy were also excluded.

### Sample size and sampling procedures

The sample size was calculated using open Epi version 3 statistical package by using 28.6% proportion of low birth weight in khat chewer groups (exposed) and 9.8% in non-khat chewers (non-exposed) from previous local study [11] and based on the assumptions of 95%, 80% power and r 1:1. The final sample size after using design effect 2 and adding 10% for loss to follow up is calculated to be 344. Dire Dawa administration, Harari regional state and Jigjiga city were purposively selected due to exposure of interest. Then, 4 hospitals (jugula, hiwot fana, dil chora and kara mara); 2 from Harari regional state (jugula and hiwot fana), one from Dire Dawa administration (dil chora) and the other one from Jigjiga city, capital of Somalia regional state (kara mara) was selected by lottery. All pregnant women being in the second trimester and early third trimester (24–28 weeks) of pregnancy visited the selected hospitals for the 1st or 2nd time during the study period was included until the required sample size of exposed and unexposed groups are fulfilled. The pregnancy follow-up contact period/time was at antenatal care appointments. This is based on the new WHO recommendations on ANC for a positive pregnancy outcome which have been started in Ethiopia [18].

### Data collection procedures and measurements

Explanatory variables data such as socio-demographic characteristics, substance use related characteristics, personal factors were collected using structured and semi-structured questionnaire at the entry to the study. The questionnaire was first prepared in English and then translated into local languages to facilitate understanding and ensure consistency during administration. In addition, anthropometric and clinical measurements were performed at entry, follow-up time and delivery (end of pregnancy) to collect the necessary data for explanatory variables.

Measurement of the status to the exposure variable (khat chewing) was performed through maternal self-report. All pregnant women to be included in the study were first assessed for khat use at the first or second prenatal visit with the use of validated questionnaire. The WHO

also recommended identification of substance use during pregnancy through interviews about substance use at antenatal care visits [19].

Weight at birth of the newborn as one of the outcome variables were measured through weighing scale to the nearest 0.1kg. Low birth weight (LBW) was defined when birth weight becomes below 2.5kg and very low birth weight (VLBW) was declared when birth weight becomes below 1.5kg [20].

Preterm birth, as the other outcome variable in the present study, was declared when the gestational age at birth of neonates became less than 37 completed weeks and after 28 completed weeks of pregnancy [21]. Gestational age was estimated in terms of weeks using maternal recall of last menstrual period (LMP). In addition, symphysis-fundal height (SFH) measurement in centimeters was performed to confirm LMP-based estimation of gestational age.

All the necessary interview and measurement data were collected by experienced health professionals working at ANC and delivery care service provision units.

## Mediators

Gestational hypertension, maternal anemia, emergency cesarean section delivery, and preterm birth were the potential mediators in the current study.

## Data quality management

Training was given for data collectors and supervisors. Data were collected by experienced health professionals working at ANC and delivery care service provision units in selected hospitals in the study area. Properly designed data collection material was developed by reviewing different literature. Strict supervision was carried out by both the supervisors and the principal investigator to check completeness and consistency. Correctly complete data was collected from data collectors by the principal investigator.

## Statistical analysis

Data analyses were performed by Statistical Package for the Social Sciences (SPSS) version 27 and Stata version 16 software. Descriptive statistics such as median, interquartile range (IQR), mean and standard deviation (SD), and frequency distribution were used to summarize the characteristics of the cohorts. Characteristic differences were determined using the chi-square test (Pearson, $p$-values tested two-sided). Survival analysis (cox proportional hazards model) and the generalized linear model for the binomial family analysis were performed to estimate the crude and adjusted relative risk and attributable risk (AR) with corresponding 95% CI. Variables with a univariate $p$-value less than or equal to 0.25 were used in the multivariable model to estimate the aRR. The relative risk with a 95% confidence interval and $p$-values was used to measure the strength of the association and to declare a statistically significant association. A statistically significant association was declared at a $p$-value $< 0.05$. Since khat chewing during pregnancy revealed a significant association with selected pregnancy outcomes, the mediation analysis [22] was considered for testing whether the supposed mediators (i.e., gestational hypertension, cesarean delivery, preterm birth, and maternal anemia) mediated the analyzed associations.

Hence, GSEM analysis was performed to examine the mediation effect of the possible mediators (i.e., gestational hypertension and cesarean delivery) on preterm birth. In the same way, GSEM analysis was also performed to examine the mediation effect of the possible mediators (i.e., gestational hypertension, cesarean delivery, maternal anemia, and preterm birth) on low birth weight. The analysis was performed using the Stata 'GSEM' command on the drop-down

menu bar. The steps of analysis were performed as follows. At first, the relationships between khat chewing during pregnancy and preterm birth and khat chewing during pregnancy and low birth weight were observed on the initial path models adjusted for potential covariates. Then, keeping in controlling of the potential covariates, the mediation analysis models were fitted to reveal the adjusted relationships between khat chewing during pregnancy, the possible consecutive mediators and preterm birth, as well as khat chewing during pregnancy, the possible mediators, and low birth weight. All the outcome variables in the present study were binary outcome variables, which have been calculated with the assumption of a Bernoulli response distribution and logit link function. Mediation has different forms. One is that mediation may be complete or partial [23, 24]. Complete mediation means the total effect of an exposure variable on the outcome variables is transmitted through one or more mediators. In this case, the exposure variable has no direct effect on the outcome variables, which means its total effect is indirect. Whereas partial mediation means an exposure variable has both direct and indirect effects on the outcome variables. Second, mediators may be single or multiple (may be consecutive) [23, 24].

In the present study, the indirect effects of khat chewing on preterm birth and low birth weight were calculated through the multiplication of regression coefficients [25]. In addition, the direct, indirect, and total effects of khat chewing on preterm birth and low birth weight have been calculated using the Stata 'nlcom' command.

## Ethical considerations

Ethical approval was obtained from the Institutional Review Board of the College of Health Sciences, Addis Ababa University. Permission was also obtained from the concerned bodies of the Dire Dawa administration, the Harari region, and the Somalia region. Moreover, informed written consent was obtained from the study participants.

## Results

### Participants classification

Out of 344 total study participants enrolled, 320 (164 non-khat chewers and 156 khat chewers) completed the follow up resulting in a loss to follow up rate of 7% (see details in Fig 1).

### Sociodemographic characteristics of the study participants

The overall mean (SD) age of participant mothers in the present study was 26.29 ±5.49 years, with the majority (38.1%) aged between 25 and 29 years old (see details in Table 1).

### Distribution of behavioral characteristics of study cohorts

Overall, 43; 31 (72.1%) chewer and 12 (27.9%) non-khat chewer study cohorts were consumers of alcohol of any type during their current pregnancy. Of them, 27(8.44%), 13 (4.1%), and 9 (2.81%) consumed beer, wine, and locally prepared alcohol (Tela), respectively. In terms of amount, 6 (15%) of the cohorts consumed 16.5grams of alcohol in a week; 14 (35%) of the study cohorts consumed 27.5-33grams of alcohol in a week; and the remaining cohorts, which are 20 (50%), consumed 49.5–55 grams of alcohol in a week. Out of the total, 18 (5.63%) of the cohorts practiced smoking tobacco products, and almost all (95%) of the study cohorts consumed coffee.

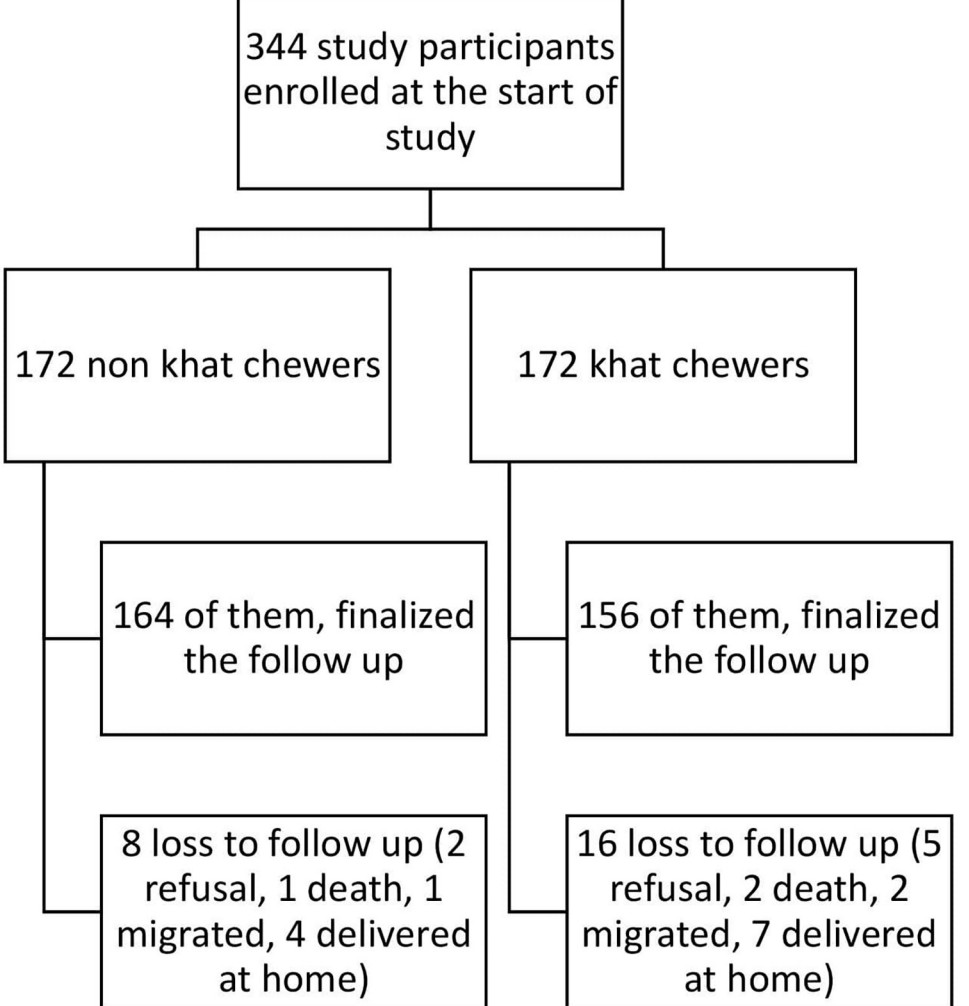

**Fig 1. Flow diagram of the study participants' involvement status in the follow up of current study conducted in Dire Dawa, Harar and Somalia selected hospitals, eastern Ethiopia, August to December 2022.**

### Pregnancy outcomes of the study cohorts

Out of the total pregnancies, 307 (95.9%) of them ended in live births, and the rest, 13 (4.1%), ended in stillbirths. The overall magnitude of congenital anomalies observed on all births, i.e., live births and stillbirths of the study cohorts, was 16 (5%), and the major birth defects identified were neural tube defects, clubfoot, orofacial clefts, malformations of the external ear, and defects of the upper and lower limbs (Table 2).

The overall magnitude of preterm births in the present study cohorts was 96 (30%); of this, 76 (79.2%) of them were from births of khat chewer cohorts. The overall low birth weight magnitude among births of the study cohorts was 127 (39.7%); of this, 94 (74%) of the low birth weight was from births of khat chewer cohorts. The overall very low birth weight magnitude in the births of the present study cohorts was 23 (7.2%) (Table 2).

### The relationship between khat chewing practices during pregnancy and selected pregnancy outcomes

As displayed in Table 3, analysis results of the Cox proportional hazards model revealed the adjusted relative risk of occurrence of preterm birth was 2.19 times higher among khat chewer

**Table 1. Distribution of sociodemographic characteristics of study participants followed from midterm of pregnancy to delivery by their khat chewing status in eastern parts of Ethiopia, 2022 (N = 320; 156 Chewers and 164 Non-chewers).**

| Characteristics | Study cohorts khat chewing status | | *p*-value |
| --- | --- | --- | --- |
| | Chewers, Frequency (%) | Non-chewers, Frequency (%) | |
| **Age of study participants (in years)** | | | |
| < = 19 | 0 | 17 (100%) | <0.001 |
| 20–24 | 48 (45.7%) | 57 (54.3%) | |
| 25–29 | 63 (51.6%) | 59 (48.4%) | |
| 30–34 | 20 (45.5%) | 24 (54.5%) | |
| > = 35 | 25 (78.1%) | 7 (21.9%) | |
| Mean (SD) age (in years) | 27.48±5.96 | 25.17±4.77 | |
| **Residence of participants** | | | |
| Urban | 50 (28.4%) | 126 (71.6%) | <0.001 |
| Rural | 106 (73.6%) | 38 (26.4%) | |
| **Ethnicity of participants** | | | |
| Oromo | 74 (50.3%) | 73 (49.7%) | 0.004 |
| Amhara | 33 (53.2%) | 29 (46.8%) | |
| Harari | 26 (48.1%) | 28 (51.9%) | |
| Somali | 23 (54.8%) | 19 (45.2%) | |
| Others | 0 | 15 (100%) | |
| **Religion of participants** | | | |
| Muslim | 110 (50.9%) | 106 (49.1%) | 0.529 |
| Orthodox | 40 (44.4%) | 50 (55.6%) | |
| Protestant | 6 (42.9%) | 8 (57.1%) | |
| **Education status of participants** | | | |
| No formal education | 54 (54.5%) | 45 (45.5%) | 0.017 |
| Primary education | 47 (58.8%) | 33 (41.3%) | |
| Secondary education | 30 (41.1%) | 43 (58.9%) | |
| Tertiary education | 25 (36.8%) | 43 (63.2%) | |
| **Participants Occupation** | | | |
| Merchant | 51 (54.8%) | 42 (45.2%) | 0.605 |
| Farmer | 43 (47.8%) | 47 (52.2%) | |
| Homemaker | 32 (47.1%) | 36 (52.9%) | |
| Employee | 23 (46%) | 27 (54%) | |
| Daily laborer | 7 (36.8%) | 12 (63.2%) | |
| **Participants marital status** | | | |
| Currently married | 129 (48%) | 140 (52%) | 0.479 |
| Divorced | 25 (55.6%) | 20 (44.4%) | |
| Widowed | 2 (33.3%) | 4 (66.7%) | |
| **Monthly HH income (ETB)** | | | |
| < = 4950.0 | 86 (52.4%) | 78 (47.6%) | 0.176 |
| >4950.0 | 70 (44.9%) | 86 (55.1%) | |

HH = household; ETB = Ethiopian birr.

study cohorts (aRR = 2.19; 95%CI 1.21–3.96) (*p*<0.005) compared to non-khat chewer study cohorts. Moreover, the occurrence of preterm birth among the current study cohorts attributed to khat chewing was 36.5% (95%CI 27.22–45.83) (*p*<0.001). In a similar analysis, the

**Table 2. Comparison of pregnancy outcomes among khat chewer and non-khat chewer maternal study cohorts in eastern Ethiopia, 2022 (N = 320; 156 Chewers and 164 Non-chewers).**

| Birth outcomes | Khat chewing behaviors of study cohorts | | p-value |
|---|---|---|---|
| | Chewers, Frequency (%) | Non-chewers, Frequency (%) | |
| **Mode of delivery** | | | |
| Normal vaginal | 69 (38.1%) | 112 (61.9%) | <0.001 |
| Instrumental | 13 (54.2%) | 11 (45.8%) | |
| Planned C/S | 14 (51.9%) | 13 (48.1%) | |
| Emergency C/S | 60 (68.2%) | 28 (31.8%) | |
| **Newborn status at birth** | | | |
| Live birth | 149 (48.5%) | 158 (51.5%) | 0.707 |
| Still birth | 7 (53.8%) | 6 (46.2%) | |
| Noticed congenital anomalies on both births | | | |
| Yes | 10 (62.5%) | 6 (37.5%) | 0.259 |
| No | 146 (48%) | 158 (52%) | |
| **Gestational age at birth (in weeks)** | | | |
| > = 37 weeks (full term birth) | 80 (35.7%) | 144 (64.3%) | <0.001 |
| <37 weeks (preterm birth) | 76 (79.2%) | 20 (20.8%) | |
| **Birth weight (in kg)** | | | |
| > = 2.5kg (normal birth weight) | 62 (32.1%) | 131 (67.9%) | <0.001 |
| <2.5kg (low birth weight) | 94 (74%) | 33 (26%) | |
| > = 1.5kg (LBW and NBW) | 134 (45.1%) | 163 (54.9%) | <0.001 |
| <1.5kg (very low birth weight) | 22 (95.7%) | 1 (4.3%) | |

C/S = cesarean section

adjusted relative risk of occurrence of low birth weight among khat chewer cohorts was 4.17 times higher (aRR = 4.17; 95%CI 2.11–8.25) ($p<0.001$) compared to non-chewer cohorts. In addition, the occurrence of low birth weight in the current study cohorts attributed to khat

**Table 3. The association between khat chewing practices during pregnancy and selected pregnancy outcomes of the study cohorts in eastern Ethiopia, 2022 (N = 320; 156 Chewers and 164 Non-chewers).**

| Birth outcome | Khat chewing characteristics of cohorts | | aRR* (95%CI) | p-value |
|---|---|---|---|---|
| | Chewers, Frequency (%) | Non-chewers, Frequency (%) | | |
| **Gestational age at birth (in weeks)** | | | | |
| Preterm birth | 76 (79.2%) | 20 (20.8%) | 2.19 (1.21–3.96) | <0.005 |
| Full term birth | 80 (35.7%) | 144 (64.3%) | 1 | |
| Birth outcome | | | | |
| **Birth weight (in kg)** | | | aRR** (95%CI) | |
| Low birth weight | 94 (74%) | 33 (26%) | 4.17 (2.11–8.25) | <0.001 |
| Normal birth weight | 62 (32.1%) | 131 (67.9%) | 1 | |
| Birth outcome | | | RR (95%CI) | |
| **Neonates' status at birth** | | | | |
| Live birth | 149 (48.5%) | 158 (51.5%) | 1 | 0.26 |
| Still birth | 7 (53.8%) | 6 (46.2%) | 1.89 (0.62–5.77) | |

\* = adjusted for residence, educational status, occupation, alcohol use, tobacco use/smoke.

\** = adjusted for maternal age, residence, ethnicity, educational status, occupation, marital status, alcohol use, tobacco smoke.

chewing was 40.1% (95%CI 30.30–49.96) ($p$<0.001). There was no statistically significant difference in the occurrence of still birth among khat chewer and non-chewer study cohorts in the present study ($p$>0.05) (Table 3).

## Mediation analysis results

The mediation analysis results of khat chewing during pregnancy and selected pregnancy outcomes were detailed in Table 4 and Figs 2 and 3. Khat chewing during pregnancy was significantly associated with preterm birth (path n, β = 0.37, $p$<0.001). More importantly significant associations were also observed between khat chewing during gestation and gestational hypertension (path k, β = 0.15, $p$ = 0.001), gestational hypertension and cesarean delivery (path l, β = 0.08, $p$<0.05), cesarean delivery and preterm birth (path m, β = 0.09, $p$<0.05). After adjusting for gestational hypertension and cesarean delivery, the regression coefficient of khat chewing during pregnancy has been decreased in size from path n, β = 0.37, $p$<0.001 to path n', β = 0.15, $p$<0.005 (Fig 2). Hence, the present study revealed that the effect of khat chewing during pregnancy on preterm birth was partially mediated by gestational hypertension and cesarean delivery.

**Table 4. The relationship between khat chewing during pregnancy, potential mediators, and selected pregnancy outcomes of the study cohorts in eastern Ethiopia, 2022 (N = 320; 156 Chewers and 164 Non-chewers): a generalized structural equation modeling analysis.**

| Model | | β*(95% CI) | $p$-value |
|---|---|---|---|
| Preterm birth | Khat chewing effect on gestational hypertension | 0.15 (0.06–0.24) | 0.001 |
| | Gestational hypertension effect on emergency cesarean delivery (c/s) | 0.08 (0.04–0.20) | <0.05 |
| | Emergency cesarean delivery effect on preterm birth | 0.09 (0.01–0.19) | <0.05 |
| | Khat use effect on preterm birth before adjustment for gestational hypertension and cesarean delivery | 0.37 (0.27–0.46) | <0.001 |
| | Khat use effect on preterm birth after adjustment for gestational hypertension and cesarean delivery | 0.15 (0.04–0.25) | <0.005 |
| Model | | β**(95% CI) | p-value |
| Low birth weight | Khat use effect on gestational hypertension | 0.15 (0.06–0.24) | 0.001 |
| | Gestational hypertension effect on emergency cesarean delivery (c/s) | 0.08 (0.04–0.29) | <0.05 |
| | Emergency cesarean delivery effect on preterm birth | 0.13 (0.02–0.25) | <0.05 |
| | Preterm birth effect on low birth weight | 0.29 (0.16–0.42) | <0.001 |
| | Khat use effect on maternal anemia | 0.19 (0.094–0.39) | <0.001 |
| | Maternal anemia on low birth weight | 0.024 (0.009–0.11) | <0.05 |
| | Khat use effect on low birth weight before adjustment for gestational hypertension, cesarean delivery, preterm birth, and maternal anemia | 0.4 (0.3–0.49) | <0.001 |
| | Khat use effect on LBW after adjustment for gestational hypertension, cesarean delivery, preterm birth, and maternal anemia | 0.2 (0.08–0.32) | <0.001 |

* = adjusted for residence, educational status, occupation, alcohol use, tobacco use/smoke.

** = adjusted for maternal age, residence, ethnicity, educational status, occupation, marital status, alcohol use, tobacco smoke.

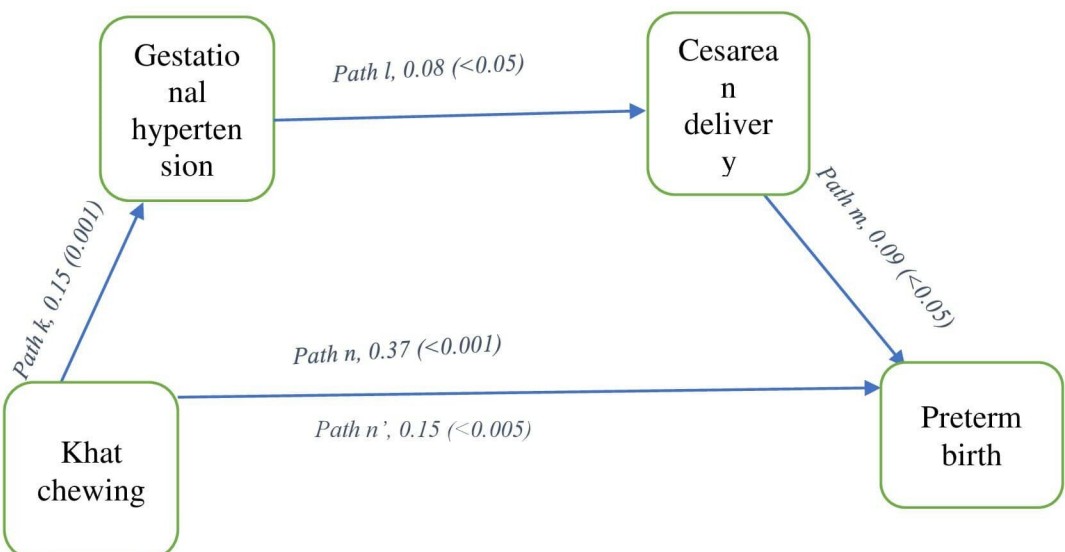

β(p-value) of path k, l and m is the indirect effects of khat chewing during pregnancy on preterm birth through gestational hypertension and cesarean delivery.

β(p-value) of path n and n' is the direct effects of khat chewing during pregnancy on preterm birth before and after adjusting for gestational hypertension and cesarean delivery respectively.

**Fig 2. Demonstrated the adjusted effect sizes of khat chewing during pregnancy on preterm birth via the consecutive mediators.**

A statistically significant association was obtained between khat chewing during pregnancy and low birth weight (path q, β = 0.4, $p<0.001$). More importantly significant associations were also observed between khat chewing during gestation and gestational hypertension (path k, β = 0.15, $p = 0.001$), gestational hypertension and cesarean delivery (path l, β = 0.08, $p<0.05$), cesarean delivery and preterm birth (path m, β = 0.13, $p<0.05$), preterm birth and low birth weight (path n, β = 0.29, $p<0.001$). At last, significant associations were also observed between khat chewing during pregnancy and maternal anemia (path o, β = 0.19, $p<0.001$), and maternal anemia and low birth weight (path p, β = 0.024, $p<0.05$). After adjusting for gestational hypertension, cesarean delivery, preterm birth and maternal anemia, the regression coefficient of khat chewing during pregnancy has been decreased in size from path q, β = 0.4, $p<0.001$ to path q', β = 0.2, $p<0.001$ (Fig 3). Hence, this finding revealed that the effect of khat chewing during pregnancy on low birth weight was partially mediated by gestational hypertension, cesarean delivery, preterm birth, and maternal anemia.

## Discussion

In the current cohort study, the magnitude as well as its relative risk of occurrence of preterm birth were significantly higher among khat chewer study participants compared to births of non-khat chewer cohorts. More importantly, the mean gestational age at birth of khat chewer cohorts (36.46±2.34weeks) is significantly lower than the mean gestational age at birth of non-khat chewer cohorts (38.10±1.66weeks). Similarly, a case control study conducted in Yemen [26] found a significantly increased risk of preterm birth among khat chewer participants compared to non-khat chewers. Even though the study finding is not in comparison of study

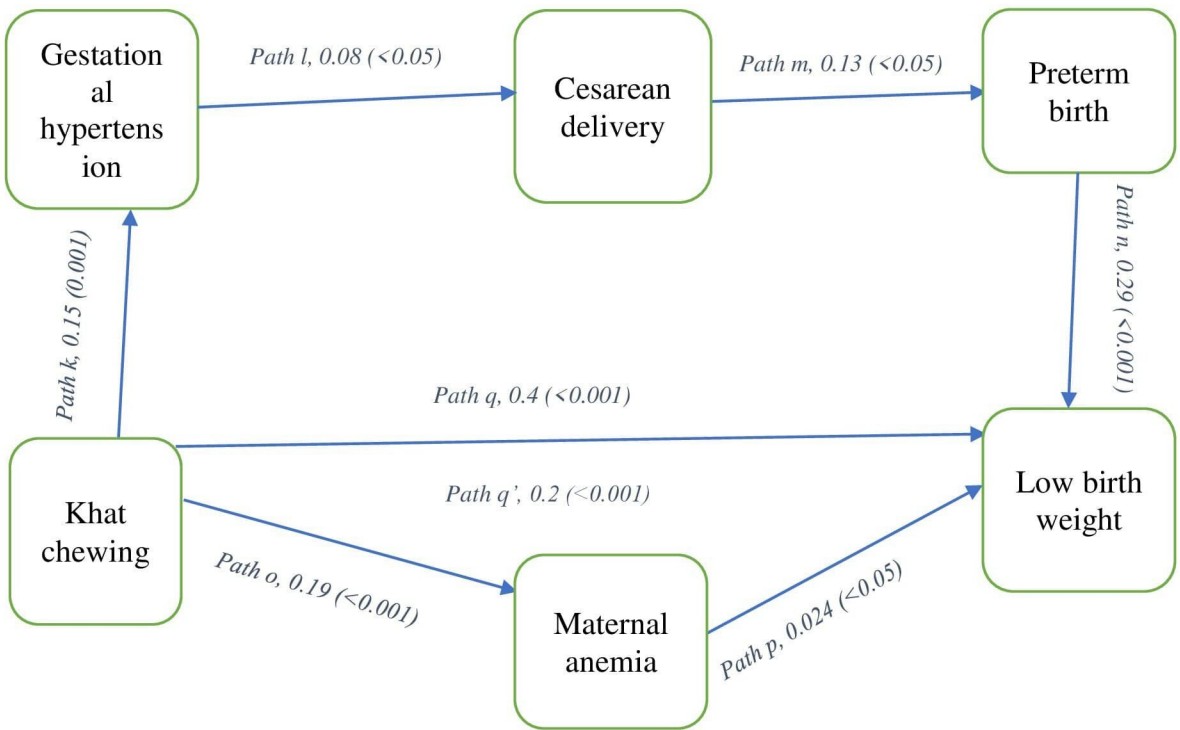

β(p-value) of path k, l, m, n, o and p is the indirect effects of khat chewing during pregnancy on low birth weight through gestational hypertension, cesarean delivery, preterm birth and maternal anemia.

β(p-value) of path q and q' is the direct effects of khat chewing during pregnancy on low birth weight before and after adjusting for gestational hypertension, cesarean delivery, preterm birth and maternal anemia respectively.

**Fig 3. Revealed the adjusted effect sizes of khat chewing during pregnancy on low birthweight through the possible mediators.**

participants with khat chewing status, a cross sectional study conducted in Jimma, Ethiopia [27] reported a 25.9% prevalence of preterm birth, and another cross sectional study conducted in Gondar, northwest Ethiopia [28] reported a 23.2% prevalence of preterm births. Moreover, a systematic review and meta-analysis study in Ethiopia [29] also reported a 37.2% pooled prevalence of preterm birth using two studies that reported preterm birth magnitude, which have been conducted on those mothers who attended ANC, and it is not known whether those mothers were using khat while pregnant. Sucha significantly increased relative risk of the occurrence of preterm birth among khat chewers may be associated with a significantly increased occurrence of preterm labor among khat chewer cohorts compared to non-chewers in the present study. A previous study reported that khat chewing during pregnancy causes labor induction [26], which may be preterm labor and then preterm birth. In addition, a significantly increased relative risk of preterm birth among khat chewers in the present study may be due to increased rates of emergency cesarean delivery compared to non-chewers. This may be due to underlined maternal and fetal factors like gestational hypertension, intrauterine

growth restriction, abruptio placenta, or pre-labor rupture of membranes (PROM). For example, khat chewing may be associated with gestational hypertension, probably due to the sympathomimetic activity of the active ingredients of khat, mainly cathinone, causing vessel constriction, which in turn leads to high blood pressure [30]. Hence, this emergency cesarean delivery may end in a preterm birth. In line with this finding, further mediation analysis of the present study found a significant association between gestational hypertension and emergency cesarean delivery and emergency cesarean delivery and preterm birth. It has been stated that fetal growth restriction and abruptio placenta may end in surgical procedures and preterm birth [31]. In addition, previous studies reported a significant association between PROM and preterm birth [32, 33].

Moreover, the overall mean gestational age at birth in this study (37.3±2.18weeks) is lower than 38.8±1.1weeks [34], 39.6±1.7weeks [35], and 39.6±1.3weeks [36] documented in northwest Nigeria, south eastern Nigeria, and Switzerland respectively. These variations in findings may be due to differences in lifestyle, culture, socioeconomic status, educational level, dietary practices, and policy of the countries.

The current cohort study found a significantly higher magnitude and increased relative risk of occurrence of low birth weight among khat chewer participants. This finding supports a case control study conducted in Yemen [26] and cross sectional studies conducted in Yemen [9, 10] and a case control study conducted in Ethiopia [11], which reported a significantly increased risk of low birth weight among births of khat chewers compared to non-khat chewer counterparts. In addition, the mean birth weight among khat chewers in the present study is significantly lower compared to non-khat chewers. In agreement with the present finding, low mean birth weight among khat chewers as compared to non-chewers has been reported in a systematic review [37]. Additionally, in line with the present finding experimental studies [6, 38] revealed decreased birth weight among khat extract administered rats compared to non-administered controls. The possible justification for this finding may be related to extrauterine and intrauterine factors. Decreased daily intake of food as demonstrated in both experimental animals [6, 39] and as well in human study [40] may be highly associated with the significant occurrence of low birth weight among khat chewers. This may be due to the fact that khat chewing may greatly influences the appetite of pregnant mothers and hence, may greatly reduce their daily intake [40] and thereby leading to maternal anemia and undernutrition which in turn has an impact on fetal birth weight. In line with this explanation further mediation analysis of the present study found a significant association between maternal anemia and low birth weight. The other is intrauterine factors. A decrease in placental blood flow, as demonstrated by an experimental study [41] and abnormal cord insertion in the human placenta [42] have been associated with a significant occurrence of low birth weight. Moreover, this significant occurrence of low birth weight among khat chewer cohorts may be due to a significant occurrence of preterm birth among them compared to the occurrence among non-chewers. In further mediation analysis, a significant association was observed between khat chewing and preterm birth, as well as between preterm birth and low birth weight. Even though no similar study exists to compare and explain the association, a previous systematic review and meta-analysis aimed at identifying determinants of low birth weight using those studies conducted on births of the general population reported preterm birth as one of the factors significantly associated with low birth weight [43].

In addition, the mean birth weight of the newborn in the current study (2717.81 ±739.29grams) is lower than 3275±469grams [34], 3400±500grams [35], 3398±484grams [44], and 3036±478grams [45] documented in north western Nigeria, south-eastern Nigeria, Ukraine and Asian populations respectively. These variations of findings may be due to differences in culture, lifestyle, dietary patterns, study area, sample size, study design, and mode of delivery.

The major limitations of the present study may be the entry period of the participants to the study which may miss those early outcomes associated with practices of chewing khat during pregnancy. This may be the main reason of the present study finding regarding no significant difference in the occurrence of birth defects among the two study groups. The other, the present study established an association between khat chewing during pregnancy and preterm birth as well as establish a relationship between chewing of khat during pregnancy and low birth weight, but the association may not be causal. Moreover, other limitation may be inability of measuring exposure status of participants using biological tests. So, relaying on self-reported data in research may have a risk of underreporting of their exposure status due to fear of prohibition or stigmatization and hence, underestimating its associated effects. But, since khat chewing practice on current study area is culturally and socially accepted phenomenon [46], the chance of misreporting is unlikely. Hence, in the interpretation of the present findings, the aforementioned limitations must be taken into consideration. But the present study has the following strengths. The first is being a prospective cohort, as it establishes temporal relationships between khat chewing during pregnancy and preterm birth and low birth weight. In addition, being a prospective cohort, the chance of missing data will be highly minimized [47]. Lastly, up to the researcher's knowledge, the present study is first in its nature, especially to determine the impacts of khat chewing during pregnancy on preterm birth and low birth weight in a prospective cohort study design in Ethiopia. It is also the first in its nature to demonstrate the mediation analysis.

## Conclusions

In the present follow-up study, the overall magnitude of preterm birth among study cohorts was 30%, with a higher proportion among births of the khat chewer cohorts. The overall low birth weight magnitude among births of the study cohorts was 39.7%, with a higher percentage among births of the khat chewer cohorts. More importantly, further analysis of the present study found that the effect of khat chewing during pregnancy on preterm birth was partially mediated by gestational hypertension and emergency cesarean delivery. In the same way, the effect of khat chewing during gestation on low birth weight was partially mediated by gestational hypertension, emergency cesarean delivery, preterm birth, and maternal anemia. Overall, the present study revealed that khat chewing is not only a worry for the current population but also a public health concern for the generation. Hence, the health professionals working in the area should highly engage in creating awareness on the harms of chewing khat during pregnancy, emphasizing the impacts on their unborn child.

## Supporting information

**S1 Data.**
(DOCX)

**S1 File.**
(SAV)

## Acknowledgments

Moreover, the authors would also like to acknowledge the study participants for their willingness to participate in the study.

## Author Contributions

**Conceptualization:** Amsalu Taye Wondemagegn.

**Data curation:** Mekbeb Afework.

**Formal analysis:** Amsalu Taye Wondemagegn, Miressa Bekana, Yonas Bekuretsion, Mekbeb Afework.

**Funding acquisition:** Amsalu Taye Wondemagegn.

**Investigation:** Miressa Bekana, Yonas Bekuretsion, Mekbeb Afework.

**Methodology:** Amsalu Taye Wondemagegn, Miressa Bekana, Yonas Bekuretsion.

**Software:** Mekbeb Afework.

**Supervision:** Miressa Bekana, Yonas Bekuretsion, Mekbeb Afework.

**Writing – original draft:** Amsalu Taye Wondemagegn.

**Writing – review & editing:** Miressa Bekana, Yonas Bekuretsion, Mekbeb Afework.

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
