## [Decision Letter · Decision Letter 0]

12 Jun 2024

PONE-D-23-42533The Impact of Chewing Khat During Pregnancy on Selected Pregnancy Outcomes in Eastern Ethiopia: A Cohort Study with A Generalized Structural Equation Modeling Analysis Approach.PLOS ONE

Dear Dr. wondemagegn,

Thank you for submitting your manuscript to PLOS ONE. After careful consideration, we feel that it has merit but does not fully meet PLOS ONE’s publication criteria as it currently stands. Therefore, we invite you to submit a revised version of the manuscript that addresses the points raised during the review process. 

We look forward to receiving your revised manuscript.

Kind regards,

Trhas Tadesse Berhe, PhD

Academic Editor

PLOS ONE

 [Financial support for this research was obtained from Addis Ababa University and Debre Markos University.].  

3. PLOS requires an ORCID iD for the corresponding author in Editorial Manager on papers submitted after December 6th, 2016. Please ensure that you have an ORCID iD and that it is validated in Editorial Manager. To do this, go to ‘Update my Information’ (in the upper left-hand corner of the main menu), and click on the Fetch/Validate link next to the ORCID field. This will take you to the ORCID site and allow you to create a new iD or authenticate a pre-existing iD in Editorial Manager. Please see the following video for instructions on linking an ORCID iD to your Editorial Manager account: https://www.youtube.com/watch?v=_xcclfuvtxQ.

Additional Editor Comments (if provided):

Reviewers' comments:

Reviewer's Responses to Questions

**Comments to the Author**

1. Is the manuscript technically sound, and do the data support the conclusions?

Reviewer #1: Yes

Reviewer #2: Partly

2. Has the statistical analysis been performed appropriately and rigorously? 

Reviewer #1: Yes

Reviewer #2: Yes

3. Have the authors made all data underlying the findings in their manuscript fully available?

Reviewer #1: Yes

Reviewer #2: Yes

4. Is the manuscript presented in an intelligible fashion and written in standard English?

Reviewer #1: Yes

Reviewer #2: No

5. Review Comments to the Author

Reviewer #1: Review Report

The proposed study is interesting and the findings are good for the PloS one readers. However, there needs improvements before the acceptance of the manuscript. The authors are recommended to revise the manuscript according to the following instructions.

1- Add more literature related to the study and highlight the significance/novelty of the proposed study.

2- Discuss the statistical tool for the analysis of the dataset in detail.

3- Add abbreviations in a Table.

4- Highlight the findings, limitations, and merits of this study.

5- There are many grammar, spelling, and sentence mistakes, authors are directed to proofread the draft thoroughly.

6- Authors are suggested to cite some recent publications in the revised version especially, Abbas et al. (2023), and Abbas et al. (2024).

Abbas, Z., Nazir, H. Z., Riaz, M., Shi, J., and Abdisa, A. G. (2023). An unbiased function‐based Poisson adaptive EWMA control chart for monitoring range of shifts. Quality and Reliability Engineering International, 39(6), 2185-2201. https://doi.org/
https://doi.org/10.1002/qre.3320

Abbas, Z., Nazir, H. Z., Xiang, D., and Shi, J. (2024). Nonparametric adaptive cumulative sum charting scheme for monitoring process location. Quality and Reliability Engineering International, https://doi.org/10.1002/qre.3522.

Reviewer #2: Extensive notes are written in the attached document.

Generally this is a sound manuscript. The written English however is poor. On most lines there are errors or grammatical mistakes. While I have tried to correct these in some instances, the mistakes were too extensive to address each one. I would recommend the manuscript is reviewed by an English-speaking copy editor before it is re-submitted.

In addition, there are a number of methodological concerns, outlined in the attached document that need to be addressed within the manuscript.

6. PLOS authors have the option to publish the peer review history of their article (what does this mean?). If published, this will include your full peer review and any attached files.

Reviewer #1: **Yes: **Zameer Abbas

Reviewer #2: No

---

## [Author Response · Author response to Decision Letter 0]

12 Jul 2024

Point-by-point response for Reviewer #1.

The authors would like to acknowledge the editors and reviewers for the valuable comments given for the betterment of our work. We have seriously looked at the comments given. Accordingly, we have corrected the manuscript as per the suggestions and see the track changes in the revised manuscript. In addition, we have a point-by-point response for each of the reviewers as follows:

Comment: 1- Add more literature related to the study and highlight the significance/novelty of the proposed study.

Response: Thank you! Corrected as per the suggestion (see the revised).

Comment: 2- Discuss the statistical tool for the analysis of the dataset in detail.

Response: Thank you! We have stated the statistical analysis used in the methodology of the study.

Comment: 3- Add abbreviations in a Table.

Response: Thank you! Added.

Comment: 4- Highlight the findings, limitations, and merits of this study.

Response: Thank you! Corrected as per the suggestions.

Comment: 5- There are many grammar, spelling, and sentence mistakes, authors are directed to proofread the draft thoroughly.

Response: Thank you! Corrected as per the comments (see the revised manuscript).

Comment: 6- Authors are suggested to cite some recent publications in the revised version especially, Abbas et al. (2023), and Abbas et al. (2024).

Response: Thank you! Cited (see the revised).

 Thank you very much!!

Point-by-point response for Reviewer #2.

The authors would like to acknowledge the editors and reviewers for the valuable comments given for the betterment of our work. We have seriously looked at the comments given. Accordingly, we have corrected the manuscript as per the suggestions and see the track changes in the revised manuscript. In addition, we have a point-by-point response for each of the reviewers as follows:

Comment: Major issues 

The authors title this paper a cohort study, but the study data was collected over the course of five months. In essence, this is a repeated measures survey. 

Khat chewing was the exposure of interest, but the authors used self-report measures to assess this, rather than any validated or biological measure. 

The authors state that the study follow-up period was antenatal care appointments, but don’t give any sense over what time period or the average time period such appointments last. For a person unfamiliar with the maternity care provided in Ethiopia, this is unclear. 

The authors don’t say if there was a primary outcome and secondary outcomes they just mention pre-term birth and low birth weight, but how were these determined and why? Why not other outcomes? The abstract methods section does not mention any outcomes at all. 

Was any blinding used in the assessment of the women in the study?

Did the study receive ethical approval? It isn’t stated in the manuscript.

There was a huge difference in the rate of attrition between the two exposure groups (double the number of khat chewers lost to follow up compared to non-khat chewers). This will have biased results to some degree, but it is not mentioned in the results or accounted for in the analysis, or even mentioned as a limitation. 

In fact, the paper has no limitations section at all, despite there been many weaknesses. The most obvious weakness is the demographic and social differences between khat chewers and non-khat chewers. The two groups are profoundly different and while the authors try to account for the differences in their analysis, there could be substantial residual confounding explaining the findings. 

Important characteristics of the women were not measured or reported, for example did the participants in the study give birth previously? Did they have any co-morbid health conditions that could also contribute to the outcome of preterm birth and low birth weight? Did participants smoke tobacco also? Are there any other potential biases in this study. The fact none are mentioned weakens this paper considerably. 

Response: Thank you very much! We have corrected as per the comments given (see the revised manuscript). 

We have provided the explanation below regarding the design as far as we know (see the revised). We have used self-report and we have purposively selected the area as chewing in the area is socially and culturally accepted phenomenon. We have stated the limitation raised as a result of self-reporting (see the revised). We have measured preterm birth & LBW. How measured? We have stated our measurement in the method section.

The was no blinding.

We have stated from where we received ethical approval i.e. IRB, CHS, AAU & it has been stated in the last paragraph of the method section.

Limitations of the study is stated as part of discussion.

Other substance use status such as Smoking status, alcohol use status, coffee use has been measured. See the result section.

Sociodemographic variables have been adjusted in analysis. 

Thank you again!

Comment: Other issues

Abstract

Line 24: Change ‘less’ to ‘little is known’ 

Line 25: Drop the word ‘hence’ and start ‘The aim…’

Line 26: Remove the repetition of the tile and just say what the study design is. 

Line 32-33: Name the primary and secondary outcomes

Line 45: Rewrite the conclusion section to be clearer and more precise. 

Response: Thank you! Corrected as per the request (See the revised manuscript) 

Thank you! Corrected as per the comment.

Comment: Introduction

Line 63: Spelling ‘chat’ instead of ‘khat’. 

Line 62 – 65: It isn’t clear from reading this what biological processes might make chewing khat during pregnancy dangerous. What is the mechanism and how might it adversely affect women and the two outcomes you are assessing?

Response: It is possible to use both. But it is largely mentioned in literature as ‘khat’ instead of ‘chat’ and we prefer it. Thank you! We have revised the mechanism (see the revised)

Comment: Methods and Materials

Line 84: As above, it may not be accurate to call this a cohort study, but it is certainly not accurate to call it a longitudinal study. 

Response: Thank you! Probably, in our previous submission we were not mentioning our recruitment time. But we have included in the revised. Otherwise, as far as we know it is a follow up study. Thank you again.

Comment: Line 85: The study says it took place in a number of selected hospitals? How many sites?

Response: Thank you! We have mentioned the sites. See the revised.

Comment: Line 95: The study says, “Dire Dawa administration, Harari regional state and Jigjiga city were purposively selected due to exposure of interest’, but can further be said on this as it isn’t clear to the reader? The exposure is khat chewing, but what is it about these regions that relates specifically to khat chewing?

Response: Thank you! Revised as per the suggestion (see the revised manuscript)

Comment: Line 97: Please re-write to clarify as it is not clear what “for the first or second time during the study period” means. 

Other important questions remain about the methodology of the study. Were any inclusion or exclusion criteria used? Were any incentives given to participate? Is maternity care accessible in Ethiopia and do women pay for it? How many antenatal care appointments do women have and does it vary widely between women? It’s possible khat chewers (given their demographic differences) are less engaged with maternity services. It is also not stated in this study, but it is important to know, how many women were approached to participate and how many said no? Who recruited and consented the participants, were the women’s medics involved in recruitment? Was fully informed consent obtained?

Response: Thank you! First or second time meaning the ANC visit number. The visit may be the 1st time with gestational age 24-28 weeks or 2nd time visit with gestational age 24-28 weeks. 

Exclusion criteria: thank you! Added as per the comment (see the revised).

It is better to see our participants classification stated in figure 1; initially we have approached 172 chewers and 172 non chewers at time of recruitment and then 156 chewers finalized and 164 of non chewers finalized the follow up.

We the researchers with trained supervisors and collectors who are working in maternity unit were involved in recruitment.

Written informed consent were obtained from the participants and it has been stated in the last paragraph of the methodology.

Comment: Line 109: Change to “Measurement of the status to the exposure variable…”

Response: Thank you! corrected (see the revised manuscript).

Comment: Line 111: Rewrite the line beginning “The WHO…” to improve clarity, it isn’t well written at present. 

Response: Thank you! corrected (see the revised manuscript)

Comment: Line 118: Rewrite the line beginning “Preterm birth…” as it is not clear. 

Response: Corrected as per the request (See the revised manuscript).

Comment: General comment: Who collected the data? Were SFH measurements taken using case notes or by a medic or researcher?

Response: Thank you! Corrected as per the comment. (see the revised) The professionals working at ANC and delivery service provision units.

Comment: Results

Table 1: Put the participant number totals in the column headings. There is no need to add the % sign in parentheses after each percentage as the ‘(%)’ in the column heading already denotes this to the reader. 

Line 226: What was adjusted? It states what was adjusted for in the table, but it should also be stated in the text. 

Response: Thank you! Corrected as per the comment (see the revised). Sociodemographic, other substance use variables have been adjusted.

Comment: Discussion

Line 289: Use ‘two’ instead of the numeral ‘2’

Line 294: Change ‘during pregnancy cause labour induction, which may be preterm labour’ to ‘during pregnancy caused labour induction which may lead to preterm labour’ 

Line 298: Spell out acronym ‘PROM’ before use

Line 313: Change line to “This finding supports a case control…”

Response: Thank you very much! We have corrected as per the comments given (see the revised).

 Thank you very much!!

---

## [Decision Letter · Decision Letter 1]

30 Jul 2024

The Impact of Chewing Khat During Pregnancy on Selected Pregnancy Outcomes in Eastern Ethiopia: A Cohort Study with A Generalized Structural Equation Modeling Analysis Approach.

PONE-D-23-42533R1

Dear Dr. wondemagegn,

We’re pleased to inform you that your manuscript has been judged scientifically suitable for publication and will be formally accepted for publication once it meets all outstanding technical requirements.

Kind regards,

Trhas Tadesse Berhe, PhD

Academic Editor

PLOS ONE

Additional Editor Comments (optional):

Reviewers' comments:

Reviewer's Responses to Questions

**Comments to the Author**

1. If the authors have adequately addressed your comments raised in a previous round of review and you feel that this manuscript is now acceptable for publication, you may indicate that here to bypass the “Comments to the Author” section, enter your conflict of interest statement in the “Confidential to Editor” section, and submit your "Accept" recommendation.

Reviewer #1: All comments have been addressed

Reviewer #2: All comments have been addressed

2. Is the manuscript technically sound, and do the data support the conclusions?

Reviewer #1: Yes

Reviewer #2: Yes

3. Has the statistical analysis been performed appropriately and rigorously? 

Reviewer #1: Yes

Reviewer #2: Yes

4. Have the authors made all data underlying the findings in their manuscript fully available?

Reviewer #1: Yes

Reviewer #2: Yes

5. Is the manuscript presented in an intelligible fashion and written in standard English?

Reviewer #1: Yes

Reviewer #2: Yes

6. Review Comments to the Author

Reviewer #1: I have no further comments as the manuscript is in improved shape .I recommend possible acceptance of this file in this well reputed international journal

Reviewer #2: Thank you for revising the paper in line with the recommendations. One minor point that should be addressed is to rewrite the last line of the introduction in the Abstract, removing the colon in the sentence.

7. PLOS authors have the option to publish the peer review history of their article (what does this mean?). If published, this will include your full peer review and any attached files.

Reviewer #1: **Yes: **I'm myself expert

Reviewer #2: **Yes: **Nicola O'Connell

---

## [Editor Report · Acceptance letter]

1 Aug 2024

PONE-D-23-42533R1 

PLOS ONE

Dear Dr. Wondemagegn, 

I'm pleased to inform you that your manuscript has been deemed suitable for publication in PLOS ONE. Congratulations! Your manuscript is now being handed over to our production team.

Kind regards, 

on behalf of

Dr. Trhas Tadesse Berhe 

Academic Editor

PLOS ONE